# Comparative Study of Coupling Techniques in Lamb Wave Testing of Metallic and Cementitious Plates

**DOI:** 10.3390/s19194068

**Published:** 2019-09-20

**Authors:** Santiago Vázquez, Jorge Gosálbez, Ignacio Bosch, Alicia Carrión, Carles Gallardo, Jordi Payá

**Affiliations:** 1Instituto de Telecomunicaciones y Aplicaciones Multimedia (iTEAM), Universitat Politècnica de València, Camino de Vera, s/n, 46022 Valencia, Spain; jorgocas@dcom.upv.es (J.G.); igbosroi@dcom.upv.es (I.B.); alcarga4@upv.es (A.C.); cargall2@upv.es (C.G.); 2Instituto de Ciencia y Tecnología del Hormigón (ICITECH), Universitat Politècnica de València, Camino de Vera, s/n, 46022 Valencia, Spain; jjpaya@cst.upv.es

**Keywords:** non-destructive testing, ultrasound, signal processing, Lamb waves, dispersion curves, angle beam wedge transducers, immersion technique, air-coupled

## Abstract

Lamb waves have emerged as a valuable tool to examine long plate-like structures in a faster way compared to conventional bulk wave techniques, which make them attractive in non-destructive testing. However, they present a multimodal and dispersive nature, which hinders signal identification. Oblique incidence is one of the most known methods to generate and receive Lamb waves and it is applied in different experimental arrangements with different types of sensors. In this work, several setups were conducted and compared to determine the optimal ones to launch and detect ultrasonic Lamb waves, especially in non-homogeneous specimens. The chosen arrangements were contact with angle beam transducers, immersion in a water tank, localised water coupling using conical containers and air coupling. Plates of two different materials were used, stainless steel and Portland cement mortar. Theoretical and experimental dispersion curves were compared to verify the existence of Lamb modes and good correspondence was achieved.

## 1. Introduction

Conventional ultrasonic inspection techniques used in non-destructive testing (NDT) are based on bulk waves (also called body or volume waves). These procedures are time-consuming since a point-by-point scan is needed to obtain global information from a structure because they only cover the localised area below or adjacent to the transducer [1]. Bulk waves consist of longitudinal (also known as P-waves/primary waves, pressure waves or compressional waves) and transverse waves (also known as S-waves/secondary waves or shear waves). Both kinds of waves occur in solids with dimensions greater than a wavelength in the three dimensions [2,3]. Nevertheless, in non-viscous fluids, such as water or air, only longitudinal waves can propagate [1]. The particle motion of longitudinal waves is parallel to the direction of wave propagation while the particle motion of transverse waves is perpendicular to the direction of wave propagation. The remaining waves, as guided waves, are composed of a combination of longitudinal and transverse wave-particle velocity components [1].

Ultrasonic guided waves have become an important subject in NDT. They are a faster, more sensitive and more economical way of inspecting materials compared with bulk wave techniques. The main advantage is that a whole structure can be inspected globally from a single probe position [4,5]. These waves need boundaries to propagate, unlike bulk waves whose propagation is unaffected by boundaries. Depending on the type of structure with boundaries (waveguide), guided waves receive different names. If the waveguide is a half-space (a thick structure) Rayleigh waves (R-waves) are found, which propagate in the surface of the tested specimen because the wavelength is very small compared to its thickness. Stoneley and Scholte waves travel in solid-solid and liquid-solid interfaces (or half-spaces), respectively. For pipes or cylindrical rods, cylindrical guided waves appear and the guided waves that occur in plates (or multilayered plates [6,7]) are known as Lamb waves [1].

Lamb waves are frequently used in ultrasonic NDT because they can travel long distances with little energy loss [8,9,10]. The main problem with Lamb waves is that they are dispersive (their phase and group velocities depend on frequency) and multimodal (different particle motion). Their velocities and modes can be obtained from characteristic Equations and the graphical solutions of these Equations are known as dispersion curves. As frequency increases, many modes with different velocities exist in the received signal, overlapping and hindering Lamb wave analysis. To improve the interpretation of these dispersive and multimode signals, different recommendations may be followed:
A region of the phase/velocity dispersion curves where little dispersion exists (i.e., small variations of phase/group velocity when changing frequency) may be selected to reduce variations in amplitude and the shape of the sensed signal as it propagates along the plate [11,12].Modes with different group velocities may be chosen to separate them in the time domain [11,13,14].Low frequencies may be chosen to minimise the number of modes in the received signals, i.e., “working on the left side” of the dispersion curves. In fact, the best would be to excite only one mode because that would enable better damage identification since the appearance of non-excited modes could indicate the existence of defects, i.e., mode conversion [11,15,16].

The fundamental modes *A_0_* and *S_0_* are frequently used for damage detection [17,18] since, at low frequencies (until the first cut-off frequency), they are the only existing modes. Normally, the *A_0_* mode presents a larger signal magnitude than the *S_0_* mode [6]. To isolate both modes, one can work with low-frequency transducers or inspect plates with the lowest thickness possible. Different types of defects that have been detected by Lamb waves are delamination [16,19], cracks [10,20], notches [21,22] and corrosion [17,23]. Depending on the type of defect, a determined mode may be excited depending on its sensitivity to that defect. Modes that present great displacements in the thickness direction, like the *A_0_* mode, have good sensitivity to surface damage, such as surface cracks [6], while modes with great displacement in the direction of wave propagation, like the *S_0_* mode, mean good sensitivity to any defect in the thickness of the specimen, such as delamination [24]. Some procedures to detect these defects include observing the velocity shifts in the experimental dispersion curves [19], the amplitude of the received signals [3] and peaks of the frequency spectrum [5].

One of the most known methods to excite guided waves, in general, is oblique incidence (also found in the literature as the wedge method or the coincidence principle) [11,14,24,25,26]. It consists of striking a bulk wave (generally a longitudinal wave, since water and air do not support transverse waves) at a determined angle with respect to the normal in a coupling medium (Plexiglas, water, air, etc.) to generate Lamb waves in the specimen by following Snell’s law. In this way, a specific mode can be selectively excited. Traditional ultrasonic probes are usually employed in oblique incidence. Although Lamb waves can be excited and measured by normal beam excitation [9,27,28] this configuration is not as efficient as oblique incidence. However, selective excitation of Lamb wave modes can be achieved by normal beam excitation. For instance, some researchers excite different modes by changing the polarity of piezoceramic lead-zirconate-titanate (PZT) transducers normally attached on both sides of the plate [8,16]. Others modify the excitation frequency and the geometry of piezoelectric wafer active sensors (PWAS, basically a network of PZT sensors) to selectively excite Lamb wave modes (Lamb wave tuning) [10,29,30]. PZT and PWAS are widely used in Lamb wave excitation and damage detection [31,32,33].

Both kind of sensors must be strongly coupled with the inspected structure. For that, different bonded materials have been studied [34,35]. They present several advantages over the typical ultrasonic probes: they are cheaper, lighter, smaller, need less power and can be embedded in structures [31]. Nonetheless, they are fragile, could show non-linearities at certain conditions and usually generate multiple Lamb wave modes [6]. In this work conventional ultrasonic transducers with oblique incidence have been used.

Different techniques can be used to change the inclination of the transducers, such as the contact technique with angle beam wedge transducers [7,36,37], immersion coupling [3,13,38] and air coupling [19,39,40]. Angle beam wedge transducers are basically longitudinal wave transducers placed on wedges with a certain angle. These wedges are usually made with materials, such as Teflon or Plexiglas, with a bulk wave velocity lower than the velocity of the mode that you want to work with. A coupling agent (gel, petroleum jelly) must be used between the transducer-wedge interface and the wedge-plate interface to provide efficient transmission of energy to the specimen [40,41]. Nevertheless, the received signal is sensitive to the thickness of the layer of the coupling gel, the contact pressure between the transducer-wedge and the wedge-plate interfaces [5,28,40] and the reflections produced within the wedges [11]. Non-contact techniques avoid these problems and enable constant coupling and faster, easier scanning [13] of the plate with repeatable results [42] compared to the direct contact case. One non-contact technique [3,13] is immersion coupling, which consists of completely submerging the test object in a fluid tank (usually water). Longitudinal wave immersion transducers are commonly employed in this setup with an acoustic matching layer between the piezoelectric element and water (the same principle is applied to piezoelectric air-coupled transducers). In this case, leaky Lamb waves (LLW), which are formed by the leakage of energy into the surrounding medium (water, air), are measured [1,38,43,44]. Note that if the acoustic impedance (or density) of the surrounding medium is small compared to the acoustic impedance of the plate, the LLW velocities can be approximated by the Lamb wave velocities [40,45,46]. Nonetheless, the received signal is formed with the addition of the generated LLW and other undesirable signals, as the reflections from the water surface, the bottom of the tank, geometrical (or specular) reflection or the direct wave through the water [28,38,44]. These undesirable signals pollute the received waveform, making mode identification difficult. To solve these issues, some researchers place some insulating material in the bottom of the tank and between the transducers [44] or increase the separation between the plate and other surfaces (bottom of the tank, water surface, etc.) [13]. Additionally, the excited mode could suffer great attenuation due to energy leakage from the plate, especially if the separation between transmitter and receiver increases (e.g., the *A_0_* mode) [3,6,11,43,47,48]. Other researchers avoid completely immersing the plate and choose water columns [42], conical water containers [28], water balloons [49] or water wedges [14,50] as the coupling mechanism. However, immersion is not practical for testing large structures, hot structures or specimens whose surface cannot be contaminated [40,51]. In these cases, air coupling is a better solution, since it maintains the advantages from immersion testing, such as constant coupling, provide fast scanning measurements [52] and almost solves the drawbacks from the undesired contributions, except for the direct acoustic wave through the air, which can be attenuated by an acoustic barrier placed between the transducers [53,54]. Two types of air-coupled transducers can be found, capacitive and piezoelectric. The capacitive design generally offers more bandwidth than the piezoelectric ceramic elements [55,56]. Nevertheless, the received waveforms show a low signal-to-noise ratio (SNR) because of the high attenuation present in air [52,53]. Therefore, amplification in transmission/reception and averaging is needed [43,52,57,58]. Furthermore, there are modes which are difficult to excite in air because of its low surface motion, such as the *S_0_* mode [40,43,52,59,60]. On the contrary, *A_0_* is a mode considered suitable for air-coupled ultrasonic testing due to its great out-of-plane displacement [39,43].

Lamb waves have been generated and received in different materials. The most commonly used are metals [3,4,8,14,36,39,43,61], but composites [9,15,24,51,60] and cementitious materials [19,62,63] are also used. However, the excitation and detection of Lamb waves in cementitious plates is still a challenging task.

The goal of this work is to compare and assess different techniques for generating and receiving Lamb waves and determine the optimal technique considering the frequency, material and mode. For this goal, the first specimen tested was a metallic plate (a reference plate in Lamb wave testing) and the contact technique, immersion technique (with two alternatives) and air coupling were analysed. The fundamental Lamb modes were successfully generated and detected. After performing a considerable amount of experiments in the metallic plate, two techniques were chosen to test a plate of a more heterogeneous material, Portland cement mortar. Despite being a more complicated material, Lamb waves were also excited and sensed. This may be due to its small thickness compared to the conventional cementitious specimens, which are often thick.

The paper is organised as follows. In Section 1, a broad introduction to Lamb waves was described. Section 2 shows the mathematical background employed in the theoretical and experimental data. In Section 3, an explanation about the different experimental arrangements is offered along with the results, which consist of matching the experimental and theoretical dispersion curves to verify which modes were excited in every case. Finally, in Section 4, the conclusions are presented.

## 2. Mathematical Background

Lamb waves propagate in a linear, homogeneous and isotropic elastic plate with stress-free upper and lower surfaces (as if the plate was placed in vacuum [36,62]) and with lateral dimensions (length and width) far greater than the thickness [39]. They are created by the constructive interference of reflections of longitudinal and transverse waves with both plate surfaces as long as the employed wavelength is greater than the plate thickness (a possible relation from [13] is 2h≤3λ, where 2h is the total plate thickness and λ the wavelength). Depending on the particle motion with respect to the middle of the plate (see Figure 1), Lamb waves can be classified into antisymmetric and symmetric wave modes (labelled $A_{m}$ and $S_{m}$, respectively, where m=0, 1, 2, 3, … indicates the “order”). Antisymmetric modes generally present out-of-plane particle displacement (in the transverse direction) while symmetric modes predominantly have in-plane particle displacement (in the longitudinal direction) [6,13,43].

### 2.1. Theoretical Dispersion Curves

For a linear, homogenous and isotropic elastic plate with stress-free boundaries and lateral dimensions far greater than the thickness (which has a value of 2h), the characteristic Equations for guided symmetric Lamb wave motion can be expressed by Equation (1) [1,10,64,65]:(1)$$4k^{2}pq\sin\left(ph\right)\cos\left(qh\right)+{\left(k^{2}-q^{2}\right)}^{2}\sin\left(qh\right)\cos\left(ph\right)=0,$$
and for antisymmetric Lamb wave motion by Equation (2):(2)$$4k^{2}pq\sin\left(qh\right)\cos\left(ph\right)+{\left(k^{2}-q^{2}\right)}^{2}\sin\left(ph\right)\cos\left(qh\right)=0,$$
(3)$$p^{2}=\frac{\omega^{2}}{c_{L}{}^{2}}-k^{2},\;\;\;q^{2}=\frac{\omega^{2}}{c_{T}{}^{2}}-k^{2},$$
where k=ωcp is the wavenumber, ω=2πf is the angular frequency, $c_{p}$ is the phase velocity and $c_{L}$ and $c_{T}$ are longitudinal and transverse velocities of the material, respectively. Characteristic Equations (1) and (2) are also known as dispersion Equations for Lamb waves or Rayleigh-Lamb Equations [6].

The solutions of Rayleigh-Lamb Equations for the phase velocity, $c_{p}$, generally plotted as a frequency depending function, are called dispersion curves. Nonetheless, there is the option to normalize the axis of the dispersion curves with respect to plate properties, e.g., the frequency axis can be multiplied by the plate thickness and the phase velocity axis can be normalized by the transverse velocity of the plate [52,62]. The phase velocity, $c_{p}$, can be defined as the speed at which the phase of any frequency component of the wave travels [39]. An example of phase velocity dispersion curves where different modes appear is found in Figure 2.

Note that for dispersion curves of symmetric modes, there is a horizontal part where the phase velocity approaches the quasi-longitudinal wave velocity [62]. In fact, the longitudinal wave is the fastest wave that appears in plates [36]. Two modes exist for all frequencies, the fundamental modes *A_0_* and *S_0_*. At higher frequencies, both modes approach the Rayleigh wave velocity of the plate while the rest of Lamb wave modes ($A_{m}$, $S_{m}$ where m>0) approach the transverse wave velocity of the plate [14]. These higher modes have cut-off frequencies where their phase velocity tends toward infinity [1,61].

When computing the dispersion curves, the real solution of the Rayleigh-Lamb Equations is chosen to represent the propagating modes of the plate. One way to achieve that is to divide Equation (1) by *q* and Equation (2) by *p*. Then, the procedure explained in [1,6] can be performed to obtain the phase velocity dispersion curves. Once the phase velocity, $c_{p}$, is known, other parameters, such as the group velocity, $c_{g}$, the angle of incidence, θ, or the wavenumber, k, can be determined. Figure 3 shows the dispersion curves of theses specific parameters for a 1.1 mm stainless steel plate.

The group velocity, $c_{g}$, can be thought of as the speed at which the wavefront of each mode propagates [64] and is given as [18,66,67]:(4)$$c_{g}=\frac{c_{p}{}^{2}}{c_{p}-f\frac{\partial c_{p}}{\partial f}}$$

The incident angle, θ, is governed by Snell’s law [26]:(5)$$c_{p}\sin\theta =c\sin\theta_{r}$$
where *c* is the bulk longitudinal wave velocity of the coupling medium (water, air, Plexiglas, etc.) and $\theta_{r}$ is the refraction angle. As θ is the critical angle which selectively excites a Lamb wave mode with a selected phase velocity, $c_{p}$, $\theta_{r}$ must be equal to 90°. Therefore, the optimum angle of incidence, θ, is [14,40,52]:(6)$$\theta =\sin^{-1}\left(\frac{c}{c_{p}}\right)$$

It is important to choose a coupling material with $c<c_{p}$. Equation (6) is the basis of the wedge method [14,25] and the coincidence principle [36,44]. Placing the receiver at the same angle as the transmitter in a pitch-catch configuration (both transducers above the plate surface) enables efficient reception (a higher amplitude) of the excited mode, suppressing others [3,11,52,59,68]. One advantage of pitch-catch setups is that they only need access to one side of the structure, in contrast to through-transmission (also called through-thickness) setups which need double-sided access [15].

Therefore, to excite and receive a certain mode, the excitation frequency and the angle of inclination with respect to normal to the transmitter and receiver must be chosen. Incident angle dispersion curves are helpful for this task [52,69]. To identify the generated modes, theoretical and experimental dispersion curves must be overlaid [16,25,37,39,62,70].

### 2.2. Experimental Dispersion Curves

Experimental phase velocity (or wavenumber) dispersion curves can be mapped out using the two-dimensional fast Fourier transform (2D-FFT) [36]. This method requires the collection of equally spaced waveforms that can be acquired by moving the receiver (or the transmitter) along the specimen [36,39,52]. With this technique, the identification of individual modes is possible [36,65].

Another procedure to measure experimental phase velocity dispersion curves is varying the angle of inclination and locating the peaks or dips [28,38,68,71] from the frequency domain of the received Lamb waves. An automated system is usually required to vary the position of the transmitter/receiver or their angles of inclination [38,39,52].

Experimental group velocity dispersion curves can be constructed from time-frequency representations (TFR) of the received signals, such as the short-time Fourier transform (STFT), the wavelet transform (WT), the scalogram, the Wigner-Ville distribution or the chirplet transform [72,73,74,75]. These techniques require only a single received waveform, which is more practical and less time-consuming [72] than the methods explained to obtain phase velocity dispersion curves (although, group velocities can also be obtained from moving the transmitter/receiver [38,39]). However, TFR presents problems due to the Heisenberg uncertainty principle [72]. In this work, the spectrogram, the graphical display from the STFT analysis that is widely used in Lamb wave studies, was applied to obtain experimental group velocity dispersion curves, which are very useful for identifying guided wave modes [1,16,37,70]. The STFT of a signal s(t) is equal to [21,72]:(7)$$S\left(f,t\right)=\frac{1}{2\pi }\int_{-\infty }^{+\infty }s\left(\tau \right)h\left(\tau -t\right)e^{-j2\pi f\tau }d\tau ,$$
where h(t) is the window function. The spectrogram is the energy density spectrum of a STFT and it is defined as:(8)$$E\left(f,t\right)={\left|S\left(f,t\right)\right|}^{2}.$$

If broadband signal excitation is used [8,16,61], several modes can be excited in a wide frequency range, which can include dispersive regions and non-dispersive regions, and only one spectrogram would be necessary. In this work, different sinusoidal tone burst signals (which produce narrowband excitation) varying in frequency [39,44,47,70,76] were launched in the tested plate and then every received spectrogram was combined into one [37] (hereafter, the combined spectrogram). The mathematical model of the transmitted sinusoidal tone burst signals is shown in Equation (9) [77]:(9)xn(t)=A·sin(2πfnt)·rect(t−Nc2fnNcfn),
where A is the signal amplitude, $f_{n}$ is the driving frequency, $N_{c}$ the number of cycles and rect(·) is the rectangular function.

Five [11,36,39,76] and ten cycles [14,58,78] are a reasonable number of cycles for burst signals. Furthermore, windowing the emitted signals can reduce side lobes to avoid exciting other modes [11,40,79] in addition to removing the ringing effects of the transducers [80].

The group velocity, $c_{g}$, can be obtained from the spectrogram. This velocity is also defined as the quotient of the propagation distance, d, in the plate and the flight time corresponding to ultrasound travel through the plate [60,81]. From the time axis, t, of the spectrogram, the experimental group velocity can be calculated as [3,13,37,70]:(10)$$c_{g}=\frac{d}{t-t_{c}},$$
where $t_{c}$ is the approximate delay suffered in the coupling medium. This delay can be obtained from the following relationships:(11)$$t_{c}=t_{c1}+t_{c2}=\frac{d_{c1}}{c}+\frac{d_{c2}}{c}=\frac{d_{c}}{c},\;$$
where $t_{c1}$ and $t_{c2}$ are the flight times for the ultrasound travelling in the coupling media from the transmitter to the plate and from the plate to the receiver, respectively, $d_{c1}$ and $d_{c2}$ are the propagation paths (transmitter-plate and plate-receiver, respectively), $d_{c}$ is the addition of both propagation paths and c is the bulk wave velocity in the coupling material, as was shown before. In this study, the three distances, d, $d_{c1}$ and $d_{c2}$, are measured using a ruler. Wherever possible, it is recommended that these distances have a value equal to or greater than the near field distance to work in the far-field region [14,53,68,82,83]. The near field distance, N, is given as [1,53]:(12)$$N=\frac{D^{2}f}{4c},$$
where D is the transducer diameter.

This recommendation is due to the near field or Fresnel region, where great ultrasound pressure fluctuations occur, hindering defect detection; while in the far-field or Fraunhofer region, the ultrasound pressure gradually tends toward zero [48,53]. If the transducers are placed close to the plate, undesirable reflections can appear between the transducer face and the specimen [59,84]. However, some researchers ignore the near field effects and adjust the distances according to the received amplitude [28] or the signal fidelity and repeatability of the experiment [3], obtaining good results. One solution to avoid distance calculations is to employ phase velocity measurements [28,36,38].

### 2.3. System Sensitivity Curves

In every arrangement, in addition to the inspected plate, the transducers and the coupling medium influence the magnitude of the received signals and, consequently, the frequency domain of those signals. By analysing this frequency domain, it is possible to know the frequencies where the excited mode presents a higher signal level (in other words, the frequencies where the mode is more sensitive). To this end, the system sensitivity curves (SSC) were computed. A diagram of the calculation of the combined spectrograms and the *SSC* is represented in Figure 4. The procedure is as follows:
1)The frequency sweep is performed and tone burst signals of different frequencies ($f_{1}$, $f_{2}$, …, $N_{f}$, where $N_{f}$ is the number of driving frequencies) are launched on the plate surface by means of the corresponding actuator.2)The generated Lamb waves are detected by the sensor and acquired for post-processing.3)Signal processing is applied and the spectrograms and the Fourier transforms (FT) for every sensed signal is obtained.4)From every spectrogram, a slice including the driving frequency is extracted and then it is normalised by that driving frequency. The slices are represented by rectangles with white discontinuous lines placed on the two spectrograms, $E_{1}\left(f,t\right)$ and $E_{N_{f}}\left(f,t\right)$. The combined spectrograms are constructed with $N_{f}$ slices [37,70].5)The *SSC* are built using the absolute values of the FT of the signals received at every injected frequency: $\left|Y_{1}\left(f_{1}\right)\right|$, $\left|Y_{2}\left(f_{2}\right)\right|$, …, $\left|Y_{N_{f}}\left(f_{N_{f}}\right)\right|$. These absolute values are marked by blue circles placed on the two FT, $\left|Y_{1}\right|$ and $\left|Y_{N_{f}}\right|$. The maximum in the FT should appear in the excitation frequency, although there are cases where this is not fulfilled. These curves act as a filter to highlight which part of the combined spectrogram can be considered in terms of SNR. In the diagram, the *SSC* presents more signal level between $f_{min}$ and $f_{max}$. Therefore, the combined spectrogram is analysed in that frequency range, i.e., between the two vertical black discontinuous lines that connects both combined spectrogram and *SSC*.
(13)$$SSC\left(f_{n}\right)=\left|Y_{n}\left(f_{n}\right)\right|,\;\;\;\;n=1,\;2,\ldots ,N_{f},$$

## 3. Experimental

Different setups were exposed for exciting and detecting Lamb waves in two plates of different materials. The transducers were placed in a pitch-catch configuration (see Figure 5 for instance) and they are mounted in holders produced by 3D printer. One transducer acts as an actuator (left) and the other as a sensor (right). To approximate the stress-free boundary conditions of the plates, a piece of insulating material (expanded polystyrene (EPS)) was placed under them [16,66,85]. A testing system was built to control the inclination and distances of the actuator/sensor. That inclination was measured using an angle level composed by an Arduino board (Mega 2560, Arduino, Somerville, MA, USA) and two sensors (MPU-6050, Invensense, San José, CA, USA). These sensors present an accuracy of ±0.03°. For every setup, a detailed explanation of the parameters (equipment, excitation, incidence angles, etc.) was given in the next sections and in Appendix B. Furthermore, the experimental data of every setup can be found as Appendix A. The signal generator and the oscilloscope used in the different setups were managed by a computer with a MATLAB graphical user interface developed by the group [86]. Special care was paid to avoid saturated signals by choosing a proper input voltage. The vertical range of the oscilloscope was adjusted to the maximum amplitude of the received signal to minimise quantification errors. Electrical stray coupling [87,88,89] and other undesired effects were removed in the post-processing stage. The spectrogram parameters remain constant for all cases, a 20 μs temporal Hamming window size, 75% overlap between windows and zero padding, up to 2^14^ points, was applied for every window. Theoretical group velocity dispersion curves were overlaid on the combined spectrograms (i.e., experimental group velocity dispersion curves). The lower and upper frequencies of the *SSC* (marked by black discontinuous lines) were superimposed on the figures at 90% of the maximum amplitude of the current *SSC*.

### 3.1. Materials

The tested samples were a stainless steel and a mortar plate. The main data of these samples are summarised in Table 1.

Although their widths should be several times larger than their thickness to optimize Lamb wave generation, some articles demonstrate that this condition is not strictly required [16,23,90,91]. The mortar was prepared by mixing a Spanish Portland cement (CEM I-52.5R), sand (quartz, 0.6–1.2 mm particle size) and water in the 1:3:0.35 ratio by mass. Superplasticiser was added in order to achieve appropriate workability. The fresh mortar was put in a mold, which was vibrated in a vibrating table (ToniVIB Model 5533, Toni Technik, Berlin, Germany) in order to homogenise the mortar and to eliminate the air bubbles. The specimens, after 24 h in the mold (temperature 20 °C, relative humidity > 95%), were demoulded and stored under water for 90 days. To determine mortar properties, the same mix was prepared and moulded in 40 × 40 × 160 mm^3^ specimens (according to UNE EN 196-1:2005 [92]) and cured in the same conditions. The bulk wave velocities of both stainless steel and mortar were measured experimentally using the Ultrasonic Pulse Velocity (UPV) method [93,94].

### 3.2. Techniques and Results on Stainless Steel

The techniques used in the stainless steel plate were the contact technique with angle beam probes, water coupling (immersion and conical containers) and air-coupled testing.

#### 3.2.1. Direct Contact

Both schematic and photograph of the experimental arrangement for the direct contact technique is shown in Figure 5. A pair of angle beam probes (MUBW 2N, Krautkramer, Huerth, Germany) with a 2 MHz central frequency were employed for Lamb wave excitation and detection. In these transducers, the piezoelectric element (9 mm large × 8 mm width) was embedded in a Plexiglas wedge, which has a longitudinal wave velocity of 2730 m/s (value obtained from the transducers data sheet). The angle can be changed manually from 0° to 60°. Petroleum jelly (Panreac, Darmstadt, Germany) was used as a coupling between the Plexiglas wedges and the plate. The effects of the coupling were ignored in the group velocity calculations. The actuator was excited by a ten-cycle tone burst using a programmable signal generator (33120A, Agilent Technologies, Loveland, CO, USA). The excitation frequencies were swept from 1 MHz to 3 MHz, in 20 kHz steps. The sent signals were amplified 40 dB (5660C, Panametrics, Waltham, MA, USA) and then captured using a digital oscilloscope (DPO3014, Tektronix, Shanghai, China) with a 25 MHz sampling frequency, 10,000 sampling points (a temporal interval of 400 μs) and 32 averaging.

Two incidence angles were chosen to excite and “follow” the predominant *S_0_* mode in the frequency range from 1 MHz to 3 MHz: 32° and 50° (see Figure 6). The *A_0_* mode cannot be generated in this arrangement since the maximum angle that can reach these transducers is 60°.

In Figure 7, the combined spectrograms for the two chosen angles along with the theoretical group velocity dispersion curves and system sensitivity curves (*SSC*) are represented. There is a good correspondence between the theoretical curve of the *S_0_* mode (red discontinuous curve) and the experimentally obtained dispersion curve (yellow) hot spots [1] in the frequency ranges that mark the *SSC*, from approximately 1.2 to 2.2 MHz in both angles, 32° and 50°. The *SSC* values for 32° are larger than those for 50° (note that the vertical ranges are set at the maximum *SSC* value of 32°). This makes sense according to the angle dispersion curves in this case (Figure 6) since 32° is a better angle than 50° to excite the *S_0_* mode (especially at 2 MHz, the centre frequency of the transducer).

#### 3.2.2. Immersion

Both schematic and photograph of the experimental arrangement for the immersion is shown in Figure 8. The plate is supported on small EPS blocks to reduce the effects of the reflections in the bottom of the tank [3,13]. A pair of broadband longitudinal transducers (K2SC, General Electric, Huerth, Germany), with a 2 MHz central frequency and 24 mm diameter were employed for Lamb wave excitation and detection. The longitudinal wave velocity in water is 1490 m/s [28,68]. The actuator was excited by a ten-cycle tone burst using a programmable signal generator (33120A, Agilent Technologies, Loveland, CO, USA). The excitation frequencies went from 1 MHz to 3 MHz, in 20 kHz steps. The sensed signals were amplified by 40 dB (5660C, Panametrics, Waltham, MA, USA) and then captured using a digital oscilloscope (DPO3014, Tektronix, Shanghai, China) with a 50 MHz sampling frequency and 10,000 point length (200 μs temporal interval) and averaged 32 times. In this case, a higher sampling frequency was used (which implies a lower acquisition time) to eliminate part of the direct wave through the water that appears after the excited modes.

The chosen angles for the water arrangement were 20° and 40° (see Figure 9). It was expected to excite *S_0_* mode with 20° and *A_0_* mode with 40°.

In Figure 10, the combined spectrograms for the two chosen angles along with the theoretical group velocity dispersion curves and *SSC* are represented. For the 20° spectrogram, the *S_0_* mode is detected only between 1.5 and 1.9 MHz (see that two frequency bands can be established in the 20° *SSC*, 1.9–2.3 MHz and 2.3–2.9 MHz) and there is close agreement between the *A_0_* mode and the 40° diagram from 1 to 2.8 MHz. In the 20° *SSC*, other modes appear because of the width of the transmitted ultrasonic beam and the reflections produced. Moreover, it is remarkable the higher values of the 40° *SSC* compared to those from the 20° *SSC* (for that reason, the vertical range of the 20° *SSC* was not set at the maximum value of the 40° SSC). This is attributed to a combination of existing reflections and the large energy leakage of the excited *A_0_* mode (40°) into the surrounding medium compared to the low energy leakage of the *S_0_* mode (20°), since the latter presents an in-plane displacement where almost all the energy is confined inside the plate [6].

#### 3.2.3. Conical Containers

Both schematic and photograph of the experimental arrangement for the setup of water conical containers is presented in Figure 11. These conical volumes were designed following [27,28]. They are bottomless, so the water is in contact with the plate. Putty was used around the conical containers to prevent water leakage. This arrangement helps to focus the ultrasonic beam (which translates in a smaller angular range) and reduces the reflections present in the immersion arrangement. Additionally, the maximum angle is limited by the conical container. A pair of broadband longitudinal transducers (K2SC, General Electric, Huerth, Germany) were employed for the excitation and detection of Lamb waves. The actuator was excited by a ten-cycle tone burst using a programmable signal generator (33120A, Agilent Technologies, Loveland, CO, USA) and amplified by a factor of 50 (WMA-300, Falco Systems, Amsterdam, The Netherlands). The excitation frequencies were varied from 1 MHz to 3 MHz, in 20 kHz steps. The sended signals were amplified 32 dB (AMPLUS-32, Dasel Sistemas, Madrid, Spain) and then captured using a digital oscilloscope (DPO3014, Tektronix, Shanghai, China) with a 25 MHz sampling frequency, 10,000 sampling points (400 μs temporal interval) and 32 averaging.

The angles chosen, as in the previous case, were 20° and 40°, for the same frequency range (1–3 MHz). It was expected to excite the same modes, *S_0_* with 20° and *A_0_* with 40°, since the coupling medium was not modified and, therefore, the same angle curves from Figure 9 could be used.

In Figure 12 the combined spectrograms for the two chosen angles along with the theoretical group velocity dispersion curves and *SSC* are presented. There is good agreement with the *S_0_* mode in the 20° diagram (1–2.5 MHz) and with the *A_0_* mode in the 40° diagram (1–1.9 MHz). The 20° combined spectrogram for conical containers differs from the 20° immersion setup. These differences are attributable mainly to the existing reflections mentioned in Section 1 that affect the signals sensed. The trend in water coupling is maintained, the larger the incidence angle, the higher the *SSC* values. Additionally, the 20° *SSC* reach higher frequencies (2.5 MHz) than the 40° *SSC* (1.9 MHz).

#### 3.2.4. Air Coupling

Both schematic and photograph of the experimental arrangement for the air coupling setup is presented in Figure 13. A block of EPS was placed between both actuator and sensor to absorb the direct wave through the air. Two pair of piezoelectric air-coupled transducers with a 32 mm diameter mm and central frequencies of 250 kHz and 500 kHz were employed for the excitation and detection of Lamb waves (information about these transducers can be found in [95,96]). The longitudinal wave velocity in air is 343 m/s [53,58]. The actuator was excited by a five-cycle tone burst using a programmable signal generator (33120A, Agilent Technologies, Loveland, CO, USA) and amplified by a factor of50 (WMA-300, Falco Systems, Amsterdam, The Netherlands). In air coupling, fewer cycles were used to minimise electrical stray coupling effects. The excitation frequencies were swept from 50 kHz to 550 kHz for the 250 kHz transducers and from 200 kHz to 800 kHz for the 500 kHz transducers, using 5 kHz steps in both cases. The sensed signals were amplified 40 dB (5660C, Panametrics, Waltham, MA, USA) and then captured using a digital oscilloscope (RTO 1004, Rohde & Schwartz, München, Germany) with a 25 MHz sampling frequency and 10,000 sampling points (400 μs temporal interval) and averaged 32 times.

Two incidence angles were chosen to excite and detect the *A_0_* mode with two kinds of transducers, 13.75° with 250 kHz transducers (frequency range from 50 to 550 kHz) and 10° with 500 kHz transducers (from 200 kHz to 800 kHz, see Figure 14). As can be seen, the lower the longitudinal wave velocity of the coupling medium, the lower the angular range in the dispersion curves. Due to the low central frequencies of the transducers (250 kHz and 500 kHz), only the two fundamental modes appear. Small angle increments can be attained since they are easy to manage, and no physical limitation exists, unlike in immersion and conical containers setups.

Several attempts were made to excite *S_0_* mode, but it was not possible as no signal was sensed by the oscilloscope. One reason is that *S_0_* particle motion is predominantly in-plane and it radiates much less energy to the air than the *A_0_* mode (as stated in Section 1) [40,43,59,60].

In Figure 15, the combined spectrograms for the two chosen angles along with the theoretical group velocity dispersion curves and *SSC* are represented. There is a good correspondence with the *A_0_* mode in the three diagrams for the frequency bands established for the *SSC*, from 0.17 to 0.33 MHz for the 13.75° diagram and from 0.3 to 0.65 MHz for the 10° diagram. The bandwidth and the *SSC* values of 10° are greater than those for the *SSC* of 13.75° (bear in mind that the transducers are different).

### 3.3. Techniques and Results on Mortar

To test the mortar plate, two techniques were selected, water conical containers and air-coupled transducers. The direct contact technique with the Plexiglas wedges was discarded for two reasons: 1) The central frequency of the wedge transducers (2 MHz) is very high to test an attenuative material like mortar [97] and 2) some modes are unable to propagate in mortar if their velocities are lower than the longitudinal wave velocity in Plexiglas because of the Snell’s law concept [26]. The Plexiglas speed (2730 m/s) is large in comparison to other coupling mediums (1490 m/s in water, 343 m/s in air). Normally, Teflon (1350 m/s) is employed as a wedge material to test cementitious materials [57,83]. Water coupling through conical containers enables the possibility of exciting both fundamental modes while air coupling enables a fast and comfortable measurement. Almost the same equipment was used. However, a lower sampling frequency (10 MHz) was chosen to enhance the temporal interval (1 ms), since the acquired signals in the mortar plate arrive later than those acquired in the metallic plate (i.e., the bulk wave velocities in mortar are lower than in the metallic plate).

#### 3.3.1. Conical Containers

Both schematic and photograph of the experimental arrangement for the setup of water conical containers to measure the mortar plate is shown in Figure 16. A pair of broadband longitudinal transducers (K0,5SC, General Electric) with a 0.5 MHz central frequency and 24 mm diameter were employed for the excitation and detection of Lamb waves. 

The actuator was excited by a ten-cycle tone burst using a programmable signal generator (33120A, Agilent Technologies, Loveland, CO, USA). The excitation frequencies were varied from 10 kHz to 1 MHz, in 5 kHz steps. The sensed signals were amplified by 40 dB (5660C, Panametrics, Waltham, MA, USA) and then captured using a digital oscilloscope (DPO3014, Tektronix, Shanghai, China) with a 10 MHz sampling frequency and 10,000 points length (a 1 ms temporal interval) and averaged 32 times. 

Two incidence angles, 20° and 40°, were chosen to excite the *S_0_* and the *A_0_* modes (respectively) in the frequency range from 0.01 to 1 MHz (see Figure 17).

In Figure 18, an acceptable coincidence exists between the theoretical and experimental curves. In a), the *S_0_* mode is detected between approximately 0.35 and 0.4 MHz (the bandwidth that shows the *SSC* goes from 0.25 to 0.55 MHz); in b) the *A_0_* mode is detected from approximately 0.1 to 0.3 MHz (the bandwidth goes from 0.05 to 0.45 MHz). As can be seen, new modes appear in these curves in contrast to the stainless steel dispersion curves (*A_2_*, *S_2_*), the larger the thickness, the larger the number of Lamb modes. In the 20° *SSC*, higher amplitude and lower bandwidth are observed. Also, the 20° *SSC* decays at a higher frequency (0.55 MHz) than the 40° *SSC* (0.45 MHz), as in Section 3.2.3 (conical containers setup to test the stainless steel plate), although no comparisons should be drawn since the inspected material and transducers are different.

#### 3.3.2. Air Coupling

Both schematic and photograph of the experimental arrangement for the air coupling setup is shown in Figure 19. The pair of piezoelectric air-coupled transducers with a 250 kHz central frequency used to measure the metallic plate were employed for the excitation and detection of Lamb waves since 250 kHz attenuates less than 500 kHz. 

The actuator was excited by a five-cycle tone burst using a programmable signal generator (33120A, Agilent Technologies) and amplified by a factor of 50 (WMA-300, Falco Systems, Amsterdam, The Netherlands). The excitation frequencies were swept from 50 kHz to 450 kHz, in 2 kHz steps (201 frequencies). The sensed signals were amplified 40 dB (5660C, Panametrics, Waltham, MA, USA) and then captured using a digital oscilloscope (RTO 1004, Rohde & Schwartz, München, Germany) with a 10 MHz sampling frequency and 10,000 points length (a 1 ms temporal interval) and averaged 32 times. A 10° angle of inclination was chosen to excite the *A_0_* mode in the mortar plate in the 50 to 450 kHz frequency range (see Figure 20).

In Figure 21, there is a good match between the theoretical and experimental curves. The *A_0_* mode is excited in the frequency range that shows the *SSC*, from 0.15 to 0.3 MHz.

## 4. Conclusions

The main conclusions are:
1)This contribution is a research work where different coupling techniques have been compared and analysed: the contact technique with angle beam probes, pure immersion and alternatives with water wedges, and air-coupled ultrasonic testing. First, a metallic plate was used to perform different experimentals and to achieve a solid theoretical basis. With this theoretical knowledge, a more complicated and heterogeneous material as mortar was inspected.2)Good matching between theoretical and experimental group velocity dispersion curves was done to determine which modes were generated in the metallic and mortar plates. Close agreement was achieved between theoretical and experimental data, which means that Lamb waves were excited and received successfully in every setup by choosing the same inclination angle in transmission and reception to enhance a particular Lamb mode.3)If the *A_0_* mode is needed, air-coupled ultrasonic testing is recommended as the first option. On the other hand, if *S_0_* mode is sought, water coupling using conical containers is suitable as a first alternative. Plexiglas wedges are also an option to excite the *S_0_* mode if the bulk wave velocities of the tested material are higher than the bulk wave velocity of Plexiglas.4)System sensitivity curves (*SSC*), a signal processing tool that represents the bandwidth of the whole “Lamb wave” system has been of great help to analyse the experimental dispersion curves in the proper frequencies.

As future directions, improvements in the conical containers design (as increasing the maximum incidence angles) and testing of more complicated cementitious materials with new configurations (as laser interferometry) will be analysed. With the acquired theoretical knowledge, practical applications of Lamb waves in cementitious materials are studied, as carbonation and thermal damage.

## Figures and Tables

**Figure 1 sensors-19-04068-f001:**
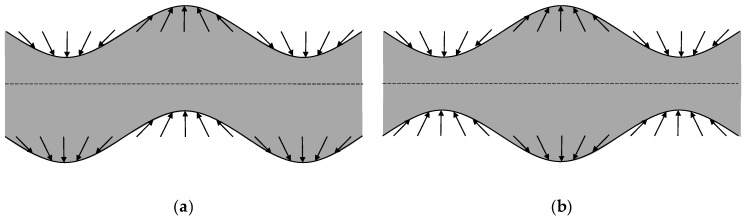
Particle displacement of (**a**) antisymmetric and (**b**) symmetric Lamb wave modes. The discontinuous line represents the middle of the plate.

**Figure 2 sensors-19-04068-f002:**
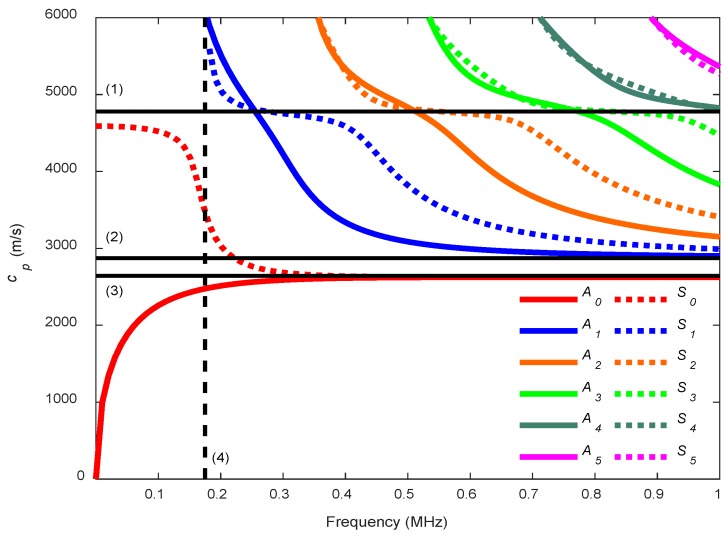
An example of Lamb wave phase velocity dispersion curves for a mortar plate ($c_{L}$ = 4779 m/s, $c_{T}\;$ = 2872 m/s, 2h = 13 mm). The three non-dispersive velocities (longitudinal (**1**), shear (**2**) and Rayleigh (**3**) wave velocities, continuous lines) and the first cut-off frequency (**4**) (discontinuous line) are highlighted. Antisymmetric modes are represented by solid lines and symmetric modes are represented by dotted lines.

**Figure 3 sensors-19-04068-f003:**
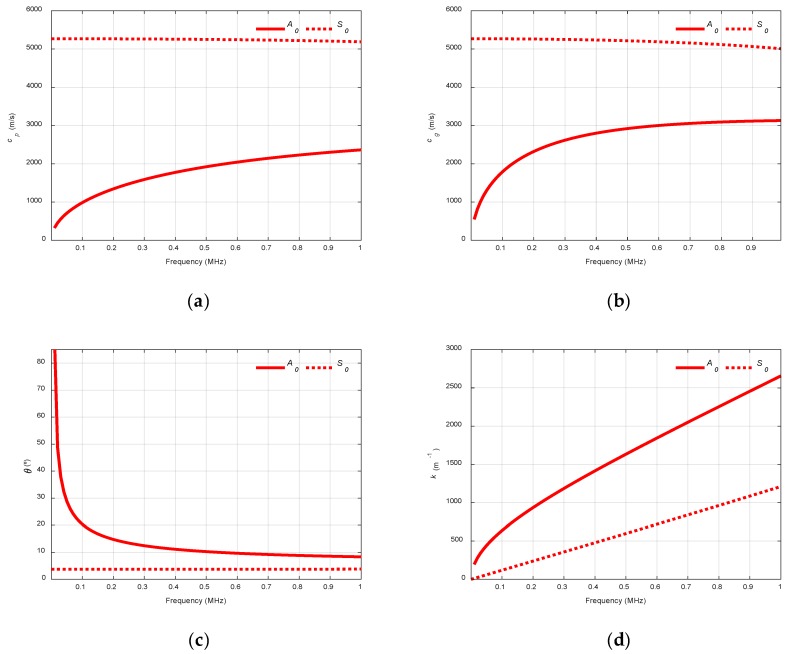
Dispersion curves for a stainless steel plate ($c_{L}$ = 5851 m/s, $c_{T}\;$ = 3056 m/s, 2h = 1.1 mm): (**a**) Phase velocity, $c_{p}$; (**b**) Group velocity, $c_{g}$; (**c**) Incident angle, θ, for air (343 m/s); (**d**) Wavenumber, k.

**Figure 4 sensors-19-04068-f004:**
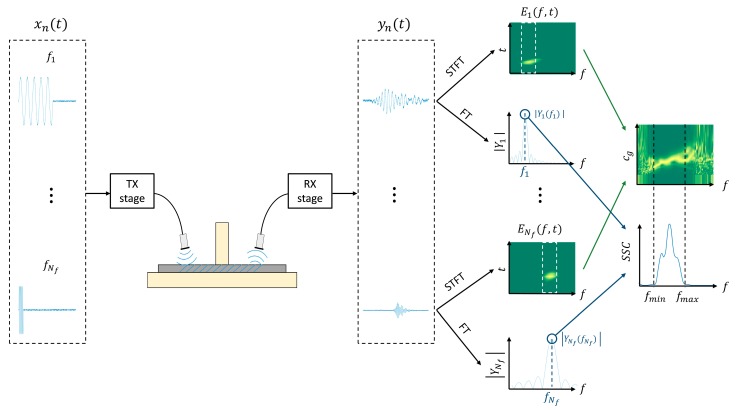
Diagram of the system sensitivity curve (*SSC*) and the combined spectrogram generation. The transmission and reception stages include part of the employed equipment (amplifiers, signal generator, oscilloscope, etc).

**Figure 5 sensors-19-04068-f005:**
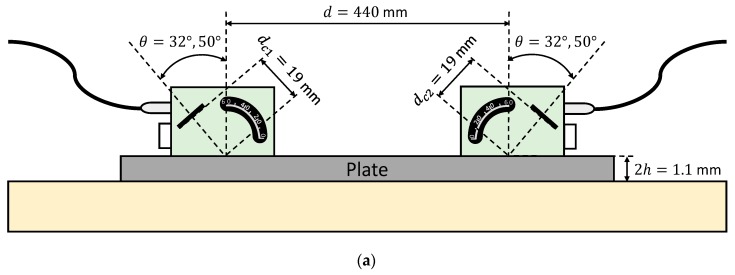
Direct contact setup with variable angle beam transducers to measure the stainless steel plate: (**a**) Schematic; (**b**) Photograph.

**Figure 6 sensors-19-04068-f006:**
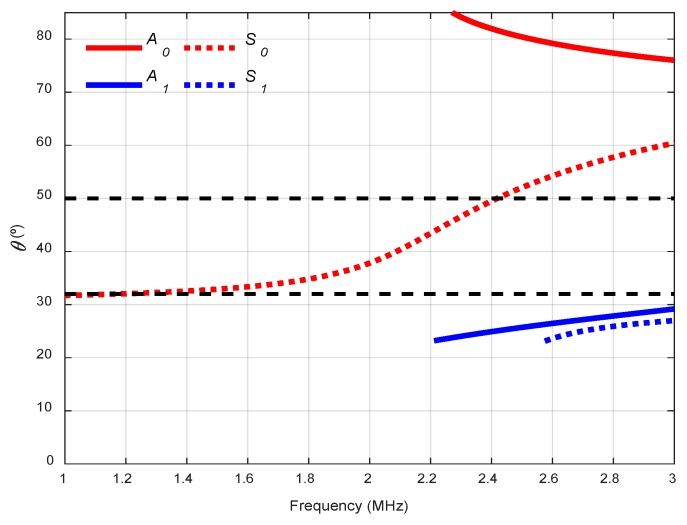
Incidence angle dispersion curves for a stainless steel plate and Plexiglas as the coupling medium (c= 2730 m/s). The chosen angles (32°, 50°) are represented by black discontinuous curves.

**Figure 7 sensors-19-04068-f007:**
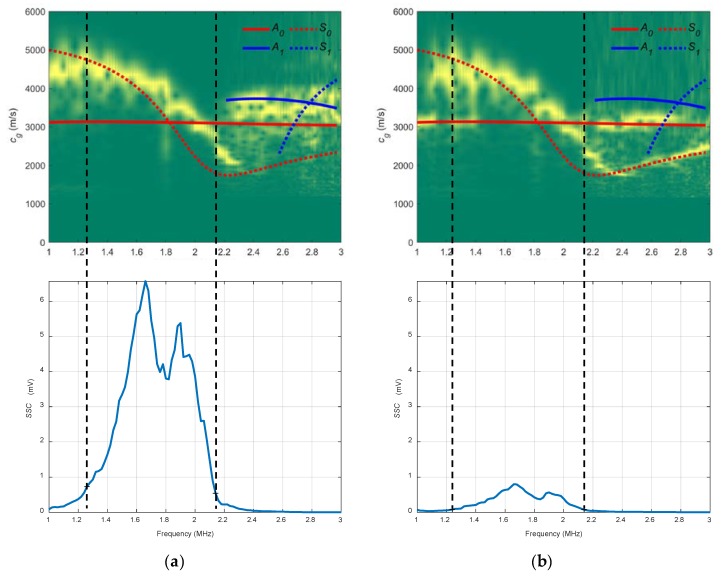
Combined spectrograms and system sensitivity curves (*SSC*) from the contact technique setup to measure the stainless steel plate: (**a**) 32°; (**b**) 50°.

**Figure 8 sensors-19-04068-f008:**
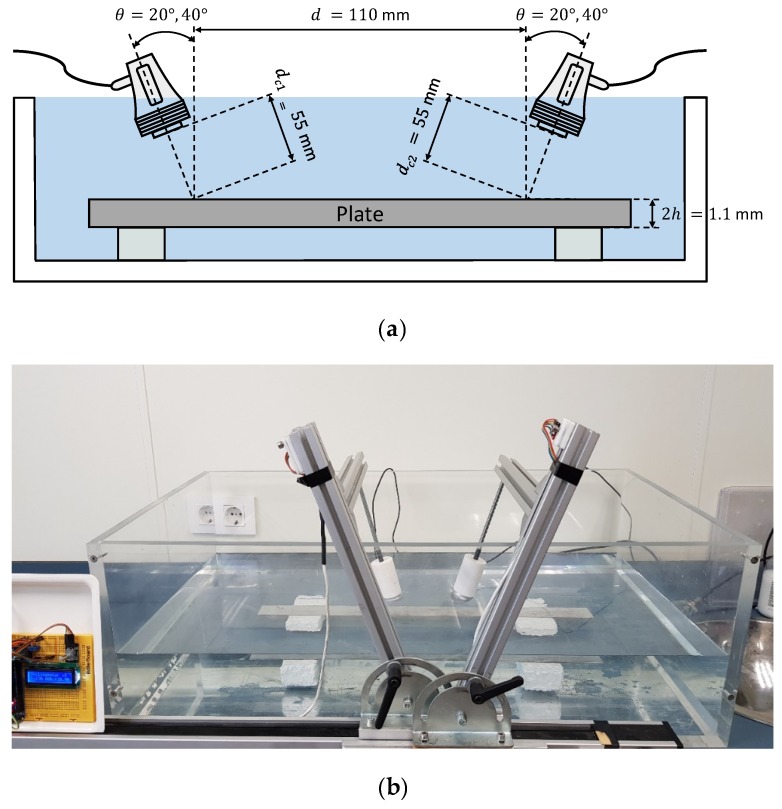
Immersion setup to measure the stainless-steel plate: (**a**) Schematic; (**b**) Photograph.

**Figure 9 sensors-19-04068-f009:**
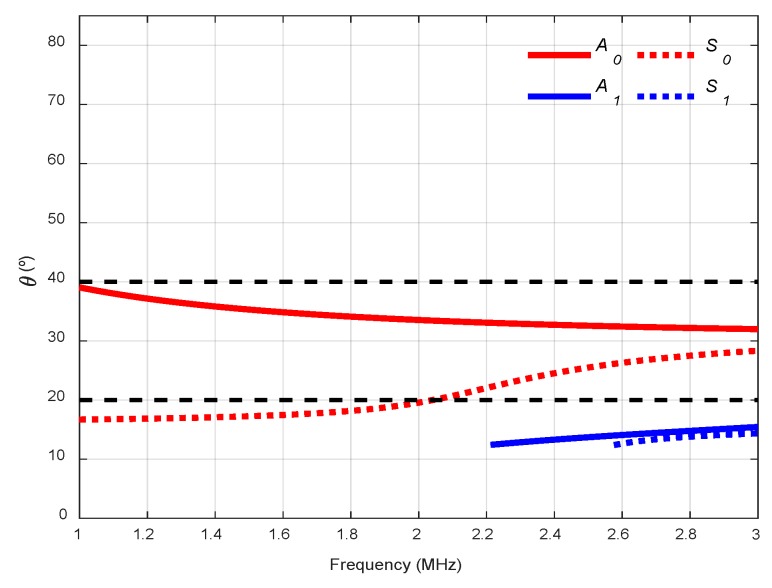
Incidence angle dispersion curves for a stainless-steel plate and water as the coupling medium (c= 1490 m/s). The chosen angles (20°, 40°) are represented by black discontinuous curves.

**Figure 10 sensors-19-04068-f010:**
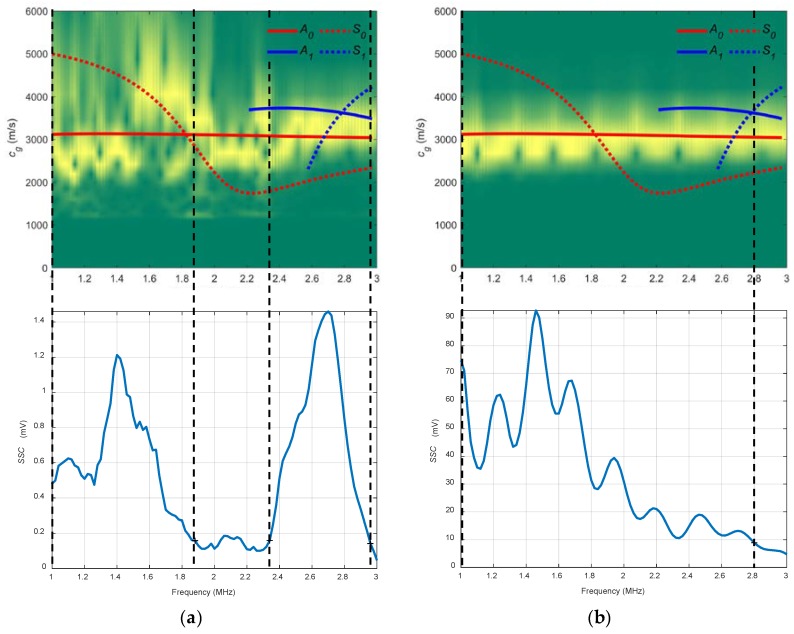
Combined spectrograms and sensitivity system curves (*SSC*) from the immersion setup to measure the stainless-steel plate: (**a**) 20°; (**b**) 40°.

**Figure 11 sensors-19-04068-f011:**
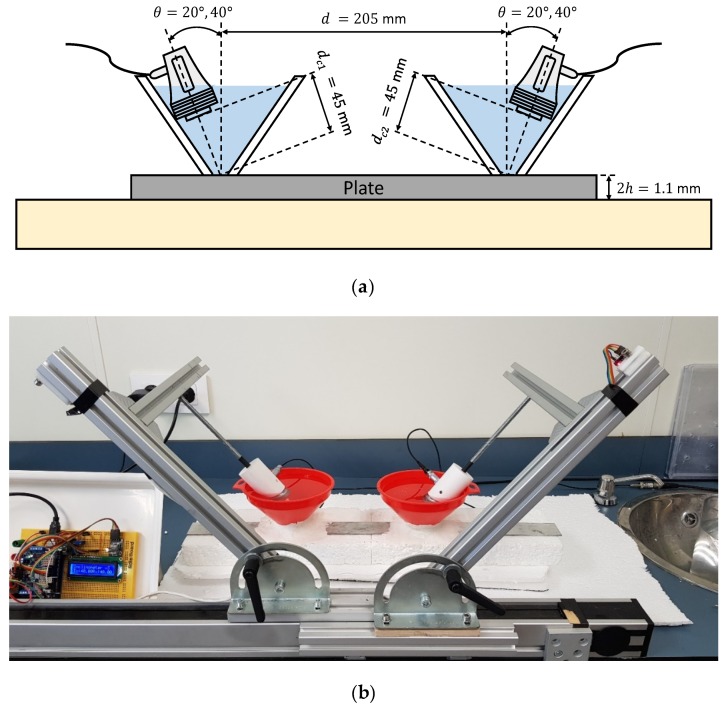
Conical containers setup to measure the stainless-steel plate: (**a**) Schematic; (**b**) Photograph.

**Figure 12 sensors-19-04068-f012:**
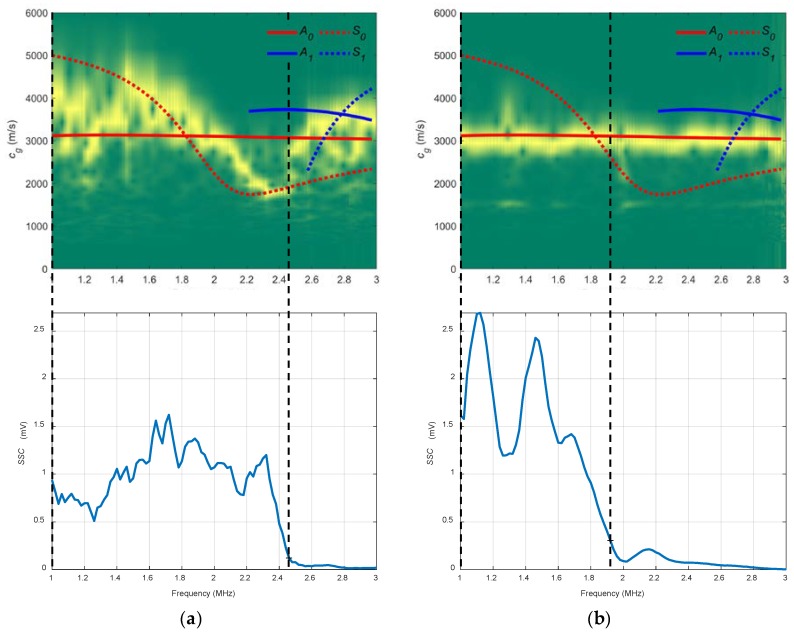
Combined spectrograms and system sensitivity curves (*SSC*) from the setup of conical containers to measure the stainless plate: (**a**) 20°; (**b**) 40°.

**Figure 13 sensors-19-04068-f013:**
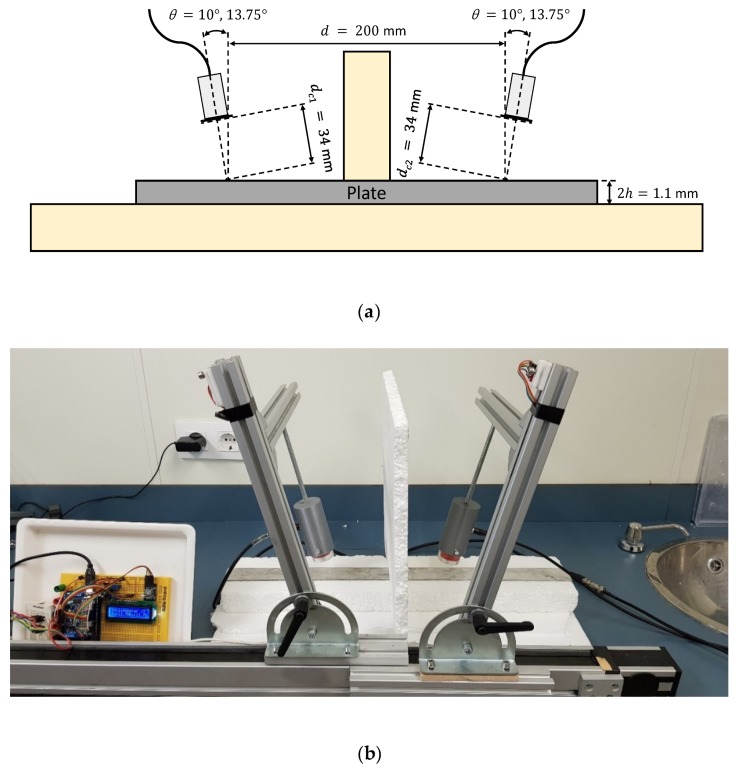
Air coupling setup.to measure the stainless-steel plate: (**a**) Schematic; (**b**) Photograph.

**Figure 14 sensors-19-04068-f014:**
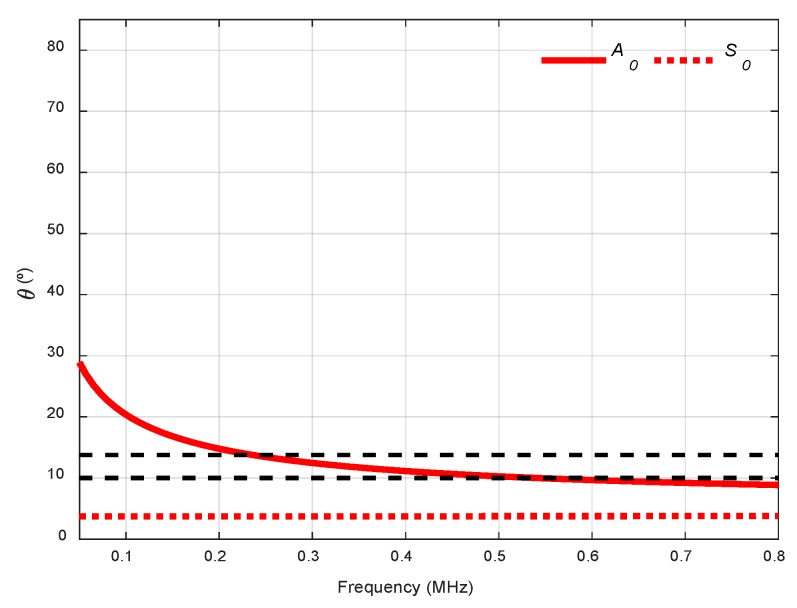
Incidence angle dispersion curves for a stainless-steel plate and air as the coupling medium (c= 343 m/s). The chosen angles (10°, 13.75°) are represented by black discontinuous curves.

**Figure 15 sensors-19-04068-f015:**
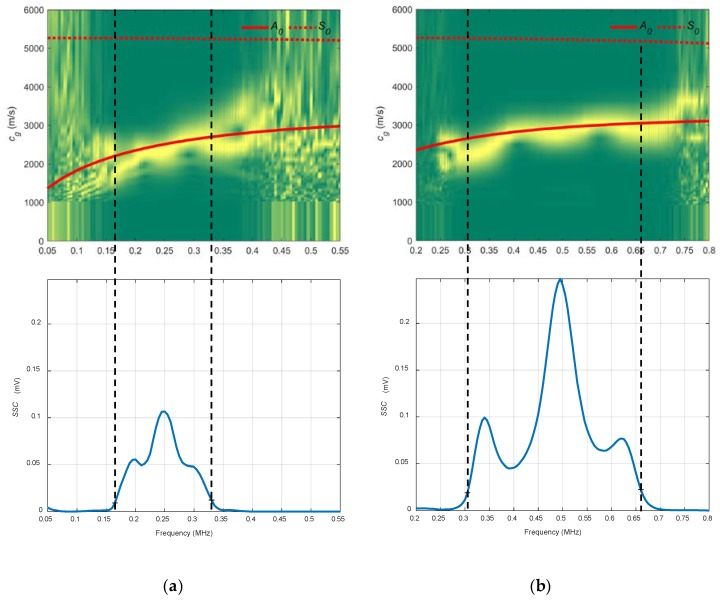
Combined spectrograms and system sensitivity curves (*SSC*) from the air coupling setup to measure the stainless-steel plate: (**a**) 13.75°, 250 kHz transducers; (**b**) 10°, 500 kHz transducers.

**Figure 16 sensors-19-04068-f016:**
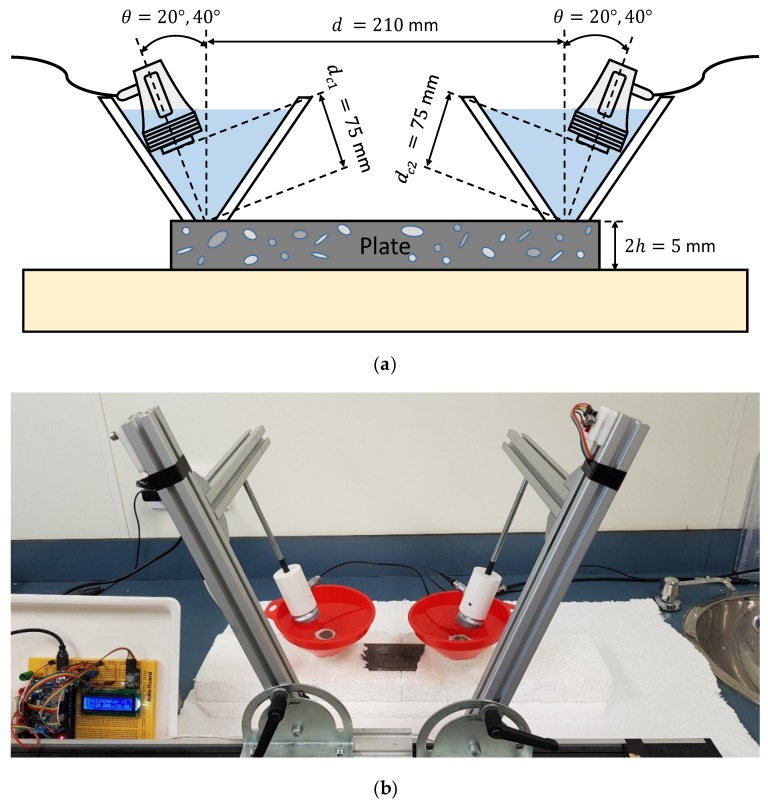
Conical containers setup to measure the mortar plate: (**a**) Schematic; (**b**) Photograph.

**Figure 17 sensors-19-04068-f017:**
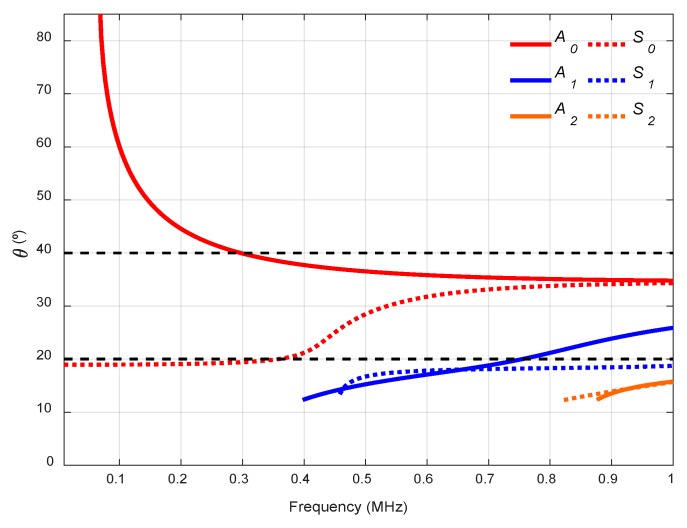
Incidence angle dispersion curves for a mortar plate and water as the coupling medium (c= 1490 m/s). The chosen angles (20°, 40°) are represented by black discontinuous curves.

**Figure 18 sensors-19-04068-f018:**
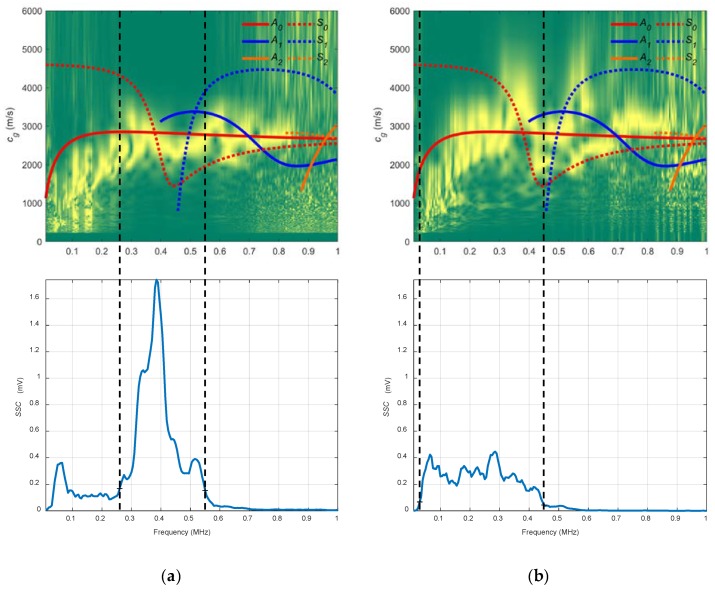
Combined spectrograms and system sensitivity curves (*SSC*) from the conical containers setup to measure the mortar plate: (**a**) 20°; (**b**) 40°.

**Figure 19 sensors-19-04068-f019:**
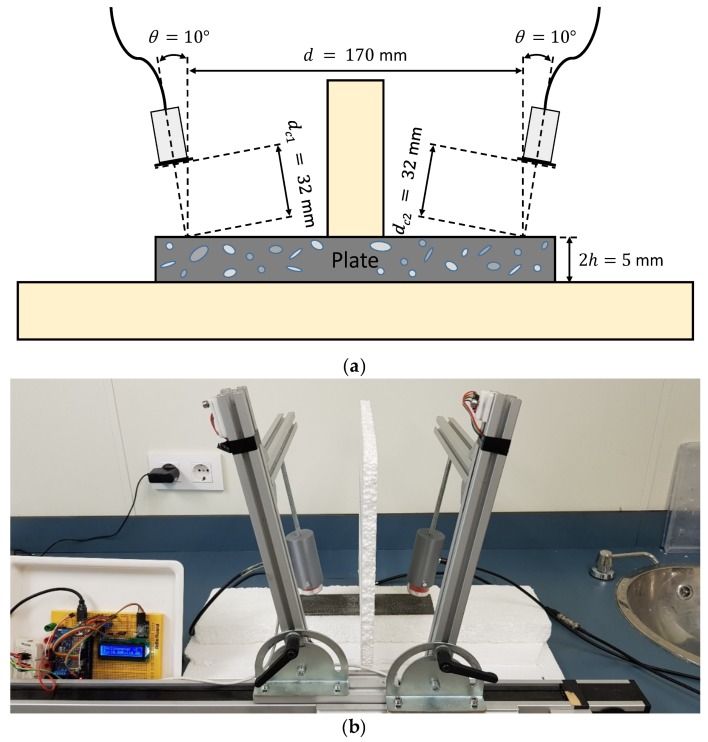
Air coupling setup to measure the mortar plate: (**a**) Schematic; (**b**) Photograph.

**Figure 20 sensors-19-04068-f020:**
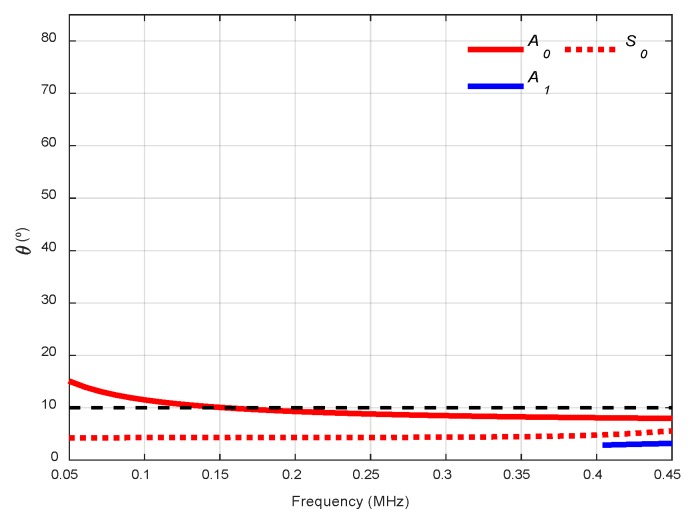
Incidence angle dispersion curves for a mortar plate and air as the coupling medium (c= 343 m/s). The chosen angle (10°) is represented by a black discontinuous curve.

**Figure 21 sensors-19-04068-f021:**
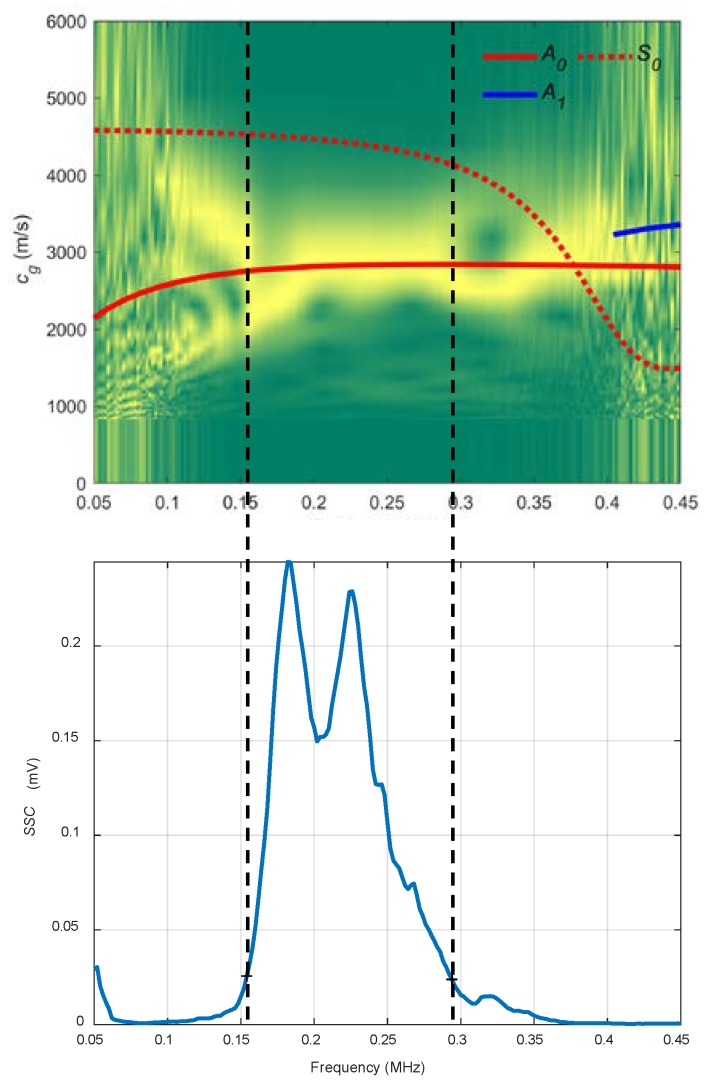
Combined spectrogram and system sensitivity curves (*SSC*) from the air coupling setup to measure the mortar plate: 10°.

**Table 1 sensors-19-04068-t001:** Specimen data.

Parameters	Stainless Steel	Mortar
Dimensions (length × width × thickness) (mm^3^)	530 × 27 × 1.1	240 × 60 × 5
Longitudinal wave velocity $c_{L}$ (m/s)	5851	4779
Transverse wave velocity $c_{T}$ (m/s)	3056	2872

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
