# Peer review of "Comparative Study of Coupling Techniques in Lamb Wave Testing of Metallic and Cementitious Plates"

_sensors, 2019, doi:10.3390/s19194068_

Round 1
Reviewer 1 Report
This paper presents a comparative study of coupling techniques in guided wave based detection. In the work, different experimental setups with different types of sensors have been tested. The paper is well organized and presented and in my opinion, it can be published after minor revisions.
To excite guided waves in structures, many researcher use PZT sensors bonded to structure surfaces. The paper only focuses on the method using ultrasonic probes. please state the author's concern or declare some limitations. Similarly, there are some paper discussing the effect of different bonding materials and methods for PZT sensors on damage detection. However, in the introduction section of the paper, the authors don't review or comment those works. In figure 2, the symmetric models should be represented by dotted lines, not by solid lines. In the conclusion section, the author can also give some future directions to develop new coupling techniques.Author Response
Please see the attachment.

Reviewer 2 Report
The paper concerns a study on Lamb wave testing of plate structures. Three different configurations of ultrasonic transducers (direct contact, immersion, conical containers) were used to obtain dispersion curves of steel and mortar plates.
The paper is well organized and easy to follow. Theoretical background and signal processing procedures used in the study are very thoroughly described, and this is the advantage of the paper. On the other side, all information provided in section Mathematical background is well established in the literature. The manuscript contains interesting experimental results; however, they mainly rely on a qualitative comparison of relation between the angle of incidence and quality of dispersion curve. The main drawback of the paper is a lack of originality.
Specific comments:
On the base of existing studies, the originality of the paper, if any, should be explained.
Section 1 is a connection of typical Introduction (containing literature review) with theoretical information about Lamb waves (e.g. Figure 1, Figure 2, lines 51-60), and in consequence, it is difficult to follow. The reviewer suggests to move some theoretical information into Section 2 or to divide Section 1 into two parts.
Figure 2 – the material and geometrical data of the plate, for which exemplary curves were plotted, should be provided.
In Section 1, some more information about excitation of Lamb waves using PWAS (Piezoelectric Wafer Active Sensors) should be provided. In particular, the reviewer suggests citing the book by Professor Giurgiutiu (Structural Health Monitoring with Piezoelectric Wafer Active Sensors) and other papers in which such sensors were used.
Section 2.3 – the diagram provided in Figure 4 should be described more precisely. The concept of the SSC should be explained.
Section 3.1 – the longitudinal and transverse wave velocities (given in Table 1) for steel should be obtained experimentally.
Section 3.1 – please specify the ingredients of the mortar – the aggregate fraction, cement class (line 320-321).
Section 3.2 – how the velocity in a Plexiglas was determined? (line 332)
Section 3.2.3 – Please provide the photograph of conical containers. Is it a commercial product or made by Authors?
Section 3 – all the equipment should be described more precisely (model, manufacturer, city and country from where equipment has been sourced), e.g. air-coupled transducers (and other apparatus).
Photographs of all experimental setups (corresponding to Figures 5, 8, 11, 13) should be added.
How the air-coupled transducers were mounted? How the incident angle of 13,75 degrees was assured (with such accuracy)?
Section Conclusions contains mainly the summary of the study, therefore it duplicates information from Section 3. It is best to only numerically and clearly list the various conclusions.
Round 2
Reviewer 1 Report
The authors answered my concerns and it is recommended to be accepted as its present form.
Reviewer 2 Report
The Authors corrected their manuscript addressing all remarks and comments. The paper was substantially improved and now it can be published in the journal.